# FXR Maintains the Intestinal Barrier and Stemness by Regulating CYP11A1-Mediated Corticosterone Synthesis in Biliary Obstruction Diseases

**DOI:** 10.3390/ijms241713494

**Published:** 2023-08-30

**Authors:** Zequn Li, Haijiang Dong, Suchen Bian, Hao Wu, Wenfeng Song, Xing Jia, Jian Chen, Xingxin Zhu, Long Zhao, Zefeng Xuan, Cheng Jin, Mengqiao Zhou, Shusen Zheng, Penghong Song

**Affiliations:** 1Division of Hepatobiliary and Pancreatic Surgery, Department of Surgery, The First Affiliated Hospital, Zhejiang University School of Medicine, Hangzhou 310003, China; 21718112@zju.edu.cn (Z.L.);; 2NHC Key Laboratory of Combined Multi-Organ Transplantation, Hangzhou 310003, China; 3Key Laboratory of the Diagnosis and Treatment of Organ Transplantation, Research Unit of Collaborative Diagnosis and Treatment for Hepatobiliary and Pancreatic Cancer, Chinese Academy of Medical Sciences (2019RU019), Hangzhou 310003, China; 4Key Laboratory of Organ Transplantation, Research Center for Diagnosis and Treatment of Hepatobiliary Diseases, Hangzhou 310003, China

**Keywords:** intestinal barrier, intestinal stemness, biliary obstruction, FXR, CYP11A1, corticosterone

## Abstract

Biliary obstruction diseases are often complicated by an impaired intestinal barrier, which aggravates liver injury. Treatment of the intestinal barrier is often neglected. To investigate the mechanism by which intestinal bile acid deficiency mediates intestinal barrier dysfunction after biliary obstruction and identify a potential therapeutic modality, we mainly used a bile duct ligation (BDL) mouse model to simulate biliary obstruction and determine the important role of the bile acid receptor FXR in maintaining intestinal barrier function and stemness. Through RNA-seq analysis of BDL and sham mouse crypts and qRT-PCR performed on intestinal epithelial-specific *Fxr* knockout (*Fxr*^ΔIEC^) and wild-type mouse crypts, we found that FXR might maintain intestinal stemness by regulating CYP11A1 expression. Given the key role of CYP11A1 during glucocorticoid production, we also found that FXR activation could promote intestinal corticosterone (CORT) synthesis by ELISA. Intestinal organoid culture showed that an FXR agonist or corticosterone increased crypt formation and organoid growth. Further animal experiments showed that corticosterone gavage treatment could maintain intestinal barrier function and stemness, decrease LPS translocation, and attenuate liver injury in BDL mice. Our study hopefully provides a new theoretical basis for the prevention of intestinal complications and alleviation of liver injury after biliary obstruction.

## 1. Introduction

Biliary obstruction is a common disease that affects a large share of the world’s population, with common causes including gallstones, tumors, inflammation, etc. [1]. It can lead to serious complications such as liver damage, bleeding, infections, malnutrition, and multiple organ failure [2]. Biliary obstruction can cause a series of pathophysiological changes in the intestinal tract, including impaired intestinal barrier function and increased permeability, and then cause bacterial displacement, aggravating other complications [3,4]. Currently, surgery is the main treatment for biliary obstruction-related diseases, while the related treatment of intestinal barrier dysfunction is often neglected. Meanwhile, targeting the intestinal epithelial barrier has proven therapeutic promise, and more research is needed to define the mechanisms [5].

The intestinal mechanical barrier consists of intestinal epithelial cells and intercellular junctions, which are important components of the intestinal barrier [5]. The intestinal epithelium is a highly dynamic structure that can complete epithelial cell renewal almost every 3–5 days. The crypt–villus structure and continuous proliferation enable the intestine to act as an absorptive organ and a protective barrier. Intestinal stem cells in intestinal crypts are the main driving cells of intestinal epithelial renewal [6]. Stem cell maintenance, regeneration, and differentiation are regulated by several factors: interleukins, Hippo signaling and metabolic cues, WNT, NOTCH, EGF, and so on [6]. Tight junctions are the main determinant of paracellular permeability and eliminate the space between intestinal cells, as observed by transmission electron microscopy [7]. The main members of intestinal tight junction proteins include the functional proteins ZO family proteins (including ZO-1, 2, 3) and structural proteins Occludin and Claudin. As a functional protein, ZO-1 mainly acts as a bridge between Occludin, Claudin, and skeletal protein F-actin [8,9]. The decrease in tight junction protein expression or structural disruption is the most important molecular manifestation of intestinal mechanical barrier disruption [1,10].

Intestinal bile acid deficiency is one of the changes in the intestinal environment in biliary obstruction-related diseases. In addition to assisting in digestion and absorption, bile acids have been proven to be an important signaling molecule linking the communication between the liver and intestine. Bile acids can regulate the function of the liver, intestine, and cells through multiple signaling pathways and even regulate the intestinal microenvironment and microecology [11,12,13]. Different bile acids can activate their corresponding bile acid receptors, mainly farnesoid X receptor (FXR or NR1H4), pregnane X receptor (PXR), vitamin D receptor (VDR), and Takeda G-protein receptor 5 (TGR5) [14,15,16,17]. Recent studies have shown that FXR is the most important receptor for bile acids to act as a signaling molecule. FXR plays an important role in bile acid homeostasis and inflammatory bowel disease, colorectal cancer, obesity, type 2 diabetes, nonalcoholic fatty liver disease, and other diseases [18]. Some studies have shown that FXR plays an important role in maintaining intestinal barrier function [19]. Obeticholic acid (OCA), an FXR-specific agonist, can partially restore intestinal barrier damage, increase permeability caused by biliary obstruction, and indirectly alleviate liver damage [20,21]. Bile acids also play an important role in the proliferation of intestinal stem cells, and lithocholic acid (LCA) has been found to promote the growth of small intestinal organoids through TGR5 [22]. However, the role of FXR in the maintenance of intestinal stemness is still controversial. One study suggests that FXR can promote the expression of tumor stem cell-related markers [23], while another suggests that the selective activation of FXR can inhibit the growth of tumor stem cells [24]. However, the specific function of FXR in common intestinal stem cells has not been clarified.

In the present study, we aimed to determine the role of bile acids and the bile acid receptor FXR in maintaining intestinal barrier function and stemness in biliary obstruction diseases. The bile duct ligation mouse model was mainly used in this study, and RNA-seq was used to discover genes regulated by FXR. Here, we investigated whether FXR activation promoted the expression of cytochrome P450 family 11 subfamily A member 1 (CYP11A1), a corticosterone synthesis rate-limiting enzyme, and increased corticosterone synthesis and secretion in the intestine. Then, corticosterone is involved in maintaining intestinal barrier function and stemness and attenuates liver injury, inflammation, and fibrosis. Our study showed a mechanism by which bile acid deficiency mediated intestinal barrier and stemness dysfunction after biliary obstruction and demonstrated the feasibility of corticosteroid treatment in biliary obstruction disease.

## 2. Results

### 2.1. Intestinal Bile Acid Deficiency Induces Intestinal Barrier and Stemness Dysfunction

Previous studies have shown that patients with biliary obstruction or animal models with BDL display intestinal barrier compromise [3,20]. However, studies on whether bile acid plays a key role that directly affects intestinal barrier function are still lacking, and the relevant mechanisms are still unclear. In this study, we performed BDL and BD experiments to simulate intestinal bile acid deficiency. Atrophied intestinal villi appeared in both BDL mice and BD mice, and a marked absence of goblet cells was observed after intestinal bile acid deficiency, indicating that the ability of the intestine to secrete mucus decreased (Figure 1A). The decline in tight junction gene (*Tjp1*, *Ocln*, and *Cldn1*) mRNA expression also illustrated intestinal barrier damage (Figure 1B). Meanwhile, similar to BDL mice, there was patchy liver inflammation damage in BD mice (Appendix A).

As intestinal cells have a high turnover rate, cellular proliferation and differentiation are remarkable features, and intestinal crypts play a key role. Then, we detected the proliferation marker PCNA, and PCNA-positive cells were significantly reduced in BDL or BD mouse intestinal crypts (Figure 1C). Intestinal epithelial renewal was dependent on intestinal stem cells. The expression of intestinal stem cell markers (*Lgr5* and *Olfm4*) was significantly decreased in the BDL mice and BD mice (Figure 1D). These results confirmed that intestinal bile acid deficiency inhibits intestinal cell proliferation and impairs intestinal stemness.

### 2.2. Activation of Fxr Maintains Intestinal Barrier Function and Stemness in BDL Mice

Bile acid receptors are the main mediators of the biological functions of bile acids. To understand the possible mechanism by which intestinal bile acid deficiency mediates intestinal barrier and stemness dysfunction after biliary obstruction, we investigated the expression of the intestinal bile acid receptors *Fxr* and *Tgr5* in the BDL, BD, and sham groups of mice. Following BDL or BD, the mRNA level of intestinal *Fxr* was significantly decreased, while the expression of *Tgr5* was not significantly different (Figure 1E). The protein expression of *Fxr* and *Tgr5* was also significantly decreased in the BDL mice intestine (Figure 1F). According to these results, we focused on the ability of *Fxr* to function in the intestinal barrier and stemness dysfunction in BDL mice.

Previous studies have shown that Fxr activation can restore the expression of the tight junction protein ZO-1 in the intestinal epithelium induced by BDL or LPS in piglets and reduce the loss of intestinal goblet cells in a mouse model of liver fibrosis [21]. Compared with wild-type mice, *Fxr* KO mice have lower ZO-1 and Claudin 1 expression and more severe LPS-induced intestinal epithelial damage [25]. In our study, we also showed that treatment with the FXR agonist OCA could reduce intestinal epithelial damage in BDL mice (Appendix A). The loss of intestinal goblet cells was mainly decreased (Figure 2A). Meanwhile, OCA treatment also reduced intestinal tight junction gene downregulation induced by BDL, including *Tjp1* (ZO-1) and *Cldn1* (Claudin 1) (Figure 2B). On the other hand, Fxr activation reduced the loss of intestinal stemness-related genes, including *Lgr5* and *Olfm4* (Figure 2D). The number of Olfm4-positive cells significantly increased in the crypts of the BDL mice after OCA treatment, indicating alleviation of intestinal stem cell loss induced by BDL (Figure 2C). OCA treatment also alleviated liver fibrosis and bile duct hyperplasia (Appendix A), which may be related to the repair of the intestinal barrier, but further experiments are needed to prove this hypothesis.

Intestinal organoids are a reliable model for studying intestinal crypt and stem cell function [26]. We cultured intestinal organoids with 50 μM CDCA, 100 nM OCA, or DMSO as a control and recorded the development of organoids in each group. Notably, increased crypt formation was observed during Fxr activation with either CDCA or OCA treatment, demonstrating that Fxr activation could promote intestinal stem cell function (Figure 2E,F). However, the mechanism by which Fxr activates intestinal stem cell function needs further experiments to be proven.

### 2.3. Fxr Affects Intestinal Corticosterone Synthesis by Regulating Cyp11a1 Expression

To further investigate the underlying mechanism of intestinal barrier and stemness dysfunction after biliary obstruction, we collected intestinal crypts from BDL or sham mice, and RNA was extracted for sequencing. RNA-Seq showed that 339 genes were upregulated and 333 genes were downregulated (Appendix A). These genes were enriched in the KEGG pathway, and the most striking enrichment was observed for metabolic pathways, including 70 genes (Appendix A). Concomitantly, the largest gene set was lipid metabolism, with 20 genes (Appendix A).

The role of bile acids as signaling agents and metabolic rate regulators is increasingly being recognized, while FXR plays a critical role [27]. To analyze whether these differential lipid metabolism-related genes were associated with altered FXR expression, we collected intestinal crypts from Fxr^ΔIEC^ and wild-type mice, and RNA was extracted. We mainly focused on the downregulated lipid metabolism-related genes, while the top five genes were selected, including *Cyp11a1*, *Agpat9*, *Asah2*, *Acacb*, and *Dgki* (Figure 3A). Validation of these five differentially expressed genes in Fxr^ΔIEC^ mice showed that *Cyp11a1* may be regulated by *Fxr*, and the expression change in the *Fxr* downstream gene *Fgf15* was also validated (Figure 3B). Deficiency of *Fxr* reduced the expression of *Cyp11a1*, while activation of Fxr had the opposite effect (Figure 3B,C). This effect was also confirmed in Caco-2 cells. When Caco-2 cells were treated with the FXR agonists CDCA or OCA, the expression of CYP11A1 was increased. FXR knockdown reduced CYP11A1 expression (Figure 3D,E).

Glucocorticoids are synthesized primarily in the cortex of the adrenal glands; the main component is cortisol in humans, while in rodents such as mice, it is corticosterone. Glucocorticoids are also synthesized in the intestinal epithelium, brain, and other tissues, and CYP11A1 is the key rate-limiting enzyme in all tissues [28,29,30]. Given the importance of CYP11A1 during steroid production, we detected the intestinal CORT level by ELISA. The results showed that BDL mice had a lower intestinal CORT level, and biliary obstruction had no effect on serum CORT levels (Appendix A). While OCA treatment significantly increased intestinal CORT levels, Fxr deficiency reduced the synthesis of CORT in the intestine (Figure 3F,G). A study had shown high expression of FXR in glucocorticoid-producing adrenocortical cells, and FXR agonist GW4064 increased plasma corticosterone levels in C57BL/6 [31]. Our data indicate that FXR could regulate intestinal corticosterone synthesis as well.

### 2.4. Corticosterone Mainly Protects Intestinal Stemness after Biliary Obstruction

To study the effects of CORT on the intestinal barrier and stemness in BDL mice, BDL mice were gavaged with CORT at a dose of 2 mg/kg (Appendix A). The intestine morphologies of sham, BDL mice, and CORT-treated BDL mice were analyzed and compared by HE staining, and the results showed that the villus shortening caused by BDL was not significantly improved after CORT gavage treatment. However, CORT treatment significantly reduced the loss of goblet cells (Figure 4A). Meanwhile, CORT-treated BDL mice showed more PCNA- or C-MYC-positive cells in intestinal crypts than BDL mice (Appendix A). The loss of Olfm4-positive stem cells in the BDL mice was significantly improved by CORT treatment (Figure 4A,B). At both the protein and mRNA levels, the expression of *Olfm4* was significantly higher after CORT treatment (Figure 4C,D). The tight junction proteins ZO-1 and Claudin 1 showed a slight improvement (Figure 4C).

Then, we analyzed goblet cell and Paneth cell compositions in the intestine in these three groups of mice by immunofluorescence staining. This result indicated that BDL caused a loss of goblet cells and Paneth cells, while CORT treatment could attenuate the effects of BDL (Appendix A). This suggested that CORT might play a role in intestinal cell differentiation. We also cultured intestinal organoids with 50 nM CORT or DMSO as a control and recorded the development of organoids in each group. Consistent with the preceding results, CORT treatment increased crypt formation and promoted organoid growth (Figure 4E,F). Furthermore, we also found that *Fxr* deficiency inhibited crypt formation, and CDCA could not improve this effect, while CORT treatment could promote crypt formation in Fxr-deficient organoids (Appendix A). This indicated that CORT synthesis might be a key step in intestinal stemness maintenance regulated by Fxr.

### 2.5. Corticosterone Reduces Liver Injury by Decreasing Intestinal Permeability

Given that intestinal barrier function is mainly reflected by intestinal permeability, we investigated it by administering 4 kDa FITC-dextran to sham, BDL mice, and CORT-treated BDL mice and found that CORT treatment could decrease intestinal permeability caused by BDL (Figure 5A). In addition, serum LPS levels were also significantly reduced by CORT treatment (Figure 5B). LPS is a major pathogenic factor derived from Gram-negative bacteria, and it may accumulate in the liver after crossing the intestinal barrier through the portal vein. Therefore, we detected the mRNA expression of *Tlr4*, *Tlr2*, and *Cd14* in the livers of the three groups of mice. These genes were mainly activated by LPS, and the results showed that CORT treatment decreased LPS-associated gene expression (Figure 5C). As shown by Sirius Red staining, CORT treatment also reduced liver damage and ductular reactions induced by BDL (Figure 5D). Moreover, CORT treatment improved liver function by decreasing ALP, AST, and ALT (Figure 5E). To confirm the attenuation of liver inflammation by CORT treatment, we measured the mRNA levels of inflammatory cytokines in the liver. *Il1a*, *Il1b*, *Il-6*, *Il-10*, and *Tnf-α* were downregulated (Figure 5F).

The results above suggested that CORT could reduce liver damage induced by BDL in mice, possibly by decreasing intestinal permeability. However, CORT is a type of corticosteroid that itself has an anti-inflammatory effect, and we cannot exclude this effect. Further experiments are needed to confirm this hypothesis.

## 3. Discussion

Although surgery is the best treatment option for biliary obstruction-related diseases, some malignant biliary obstructions due to pancreatic adenocarcinoma and cholangiocarcinoma, or intractable biliary obstructions, including primary sclerosing cholangitis and primary biliary cholangitis, are often not immediately operable or inoperable [3,10,32]. Chronic biliary obstruction impairs the intestinal barrier and increases intestinal permeability, causing endotoxemia or bacteremia, which can aggravate liver injury [33,34,35]. Therefore, targeting the intestinal barrier is a promising treatment to attenuate liver injury in these malignant biliary obstructions and intractable biliary obstructions and may prolong survival or increase the chance of surgery.

The BDL mouse model could simulate the changes in the intestinal environment and pathophysiology in biliary obstruction to a certain extent, and to determine whether intestinal barrier and stemness dysfunction after biliary obstruction are directly related to intestinal bile acid deficiency, we also performed a BD mouse model to simulate intestinal bile acid deficiency. Our work suggests that bile acids are essential for maintaining the intestinal barrier and stemness and that the bile acid receptor FXR might play a key role (Figure 1). Interestingly, FXR activation protected intestinal stem cells from BDL and promoted the development of intestinal organoids. In intestinal organoids from Fxr^ΔIEC^ mice, crypt formation was inhibited (Appendix A). Conversely, selective activation of intestinal FXR could restrict intestinal cancer stem cell proliferation, as Ting et al. reported [24]. This suggests that the function of FXR in noncancerous or cancerous tissues might be different, and the mechanism requires further experiments.

Previous studies have found that corticosterone can be synthesized in tissues other than the adrenal gland, such as the intestinal epithelium, thymus, and brain, and CYP11A1 and other synthetases can also be expressed [28,29]. However, the role of these nonadrenal corticosterone in the tissues is still unclear. In our study, decreased mRNA expression of Cyp11a1 after BDL was shown in the results of RNA-Seq in the BDL and sham mouse crypts. *Cyp11a1* expression was also downregulated in *Fxr*^ΔIEC^ mouse crypts. Meanwhile, FXR activation promoted CYP11A1 expression in vivo and in vitro and promoted intestinal corticosterone synthesis, while BDL reduced intestinal corticosterone without affecting the concentration of corticosterone in the blood (Figure 3). Therefore, we suggest that the expression of CYP11A1 and corticosterone synthesis in the intestine are regulated by FXR. Further study is needed to clarify the detailed mechanism. Corticosteroids (including corticosterone) have two roles in regulating various barrier functions or stem cell functions in vivo. Some studies believe that their role is to protect barrier function and promote stem cell proliferation and differentiation [36,37]. It has also been suggested that corticosteroids may increase barrier damage and inhibit stem cell growth [38,39,40]. In our study, corticosterone treatment significantly decreased the intestinal permeability of BDL mice, reduced the amount of FITC-dextran penetrating the intestinal barrier, reduced LPS translocation, and promoted the expression of the tight junction protein ZO-1. On the other hand, corticosterone also reduced the loss of stemness genes in the intestine of BDL mice and maintained the number of various cells in the intestine. The intestinal epithelium of the BDL mice treated with corticosterone had more goblet cells and Paneth cells than that of the BDL mice without corticosterone treatment (Appendix A). Corticosterone stimulation in vitro also promoted the growth of small intestinal organoids and promoted crypt formation in Fxr-deficient organoids (Figure 4E and Appendix A). Our results suggest that corticosterone has positive effects on intestinal barrier function and stemness.

Some clinical trials have reported that corticosteroids combined with ursodeoxycholic acid (UDCA) can effectively improve biochemical markers of disease activity caused by primary biliary cholangitis [41,42]. Our results showed that corticosterone treatment significantly reduced serum liver function indexes, liver inflammation-related indexes, and histopathological liver injury and fibrosis (Figure 5). Therefore, our study validates the feasibility of corticosteroids in the treatment of cholestatic liver injury after biliary obstruction, and the reduction in the intestinal barrier and stemness dysfunction may be one of the key factors of corticosterone protection of the liver. However, this study still has some limitations. Although corticosterone treatment was administered by gavage, the possibility that corticosterone entering the blood directly works on the liver and reduces inflammation cannot be excluded. Therefore, corticosterone attenuates liver injury in mouse models of biliary obstruction, most likely because it plays an important role in both the liver and intestine. A study reported that individuals with high serum glucocorticoid levels also inhibited hepatic FXR transcriptional activity, thereby promoting intrahepatic cholestasis in mice [43]. Thus, the dose of glucocorticoids for the treatment of biliary obstruction in the clinic needs to be further determined.

In conclusion, our findings suggest that biliary obstruction could induce intestinal barrier and stemness dysfunction, which was mainly associated with intestinal bile acid deficiency. The bile acid receptor FXR plays a key role. FXR activation promoted the expression of CYP11A1 and increased corticosterone synthesis in the intestine. Then, corticosterone is involved in maintaining intestinal barrier function and stemness. Corticosterone treatment decreased intestinal permeability, reduced harmful substances in the blood, and attenuated liver injury, inflammation, and fibrosis in the BDL mouse model. Our study showed the mechanism by which bile acid deficiency mediated intestinal barrier and stemness dysfunction after biliary obstruction and demonstrated the feasibility of glucocorticoid treatment in biliary obstruction disease. It hopefully provides a new theoretical basis for the prevention and treatment of intestinal complications and alleviation of liver injury after biliary obstruction.

## 4. Materials and Methods

### 4.1. Animals and Models

The animal care and experiments were approved by the Tab of Animal Experimental Ethical Inspection of the First Affiliated Hospital, Zhejiang University School of Medicine. Male C57BL/6 mice, male *Nr1h4*^fl/fl^; Vil-Cre mice with intestinal epithelial-specific *Fxr* knockout (*Fxr*^ΔIEC^), and *Nr1h4*^fl/fl^ mice without Cre expression (WT) were used in this study, supplied by Cyagen Biosciences (Suzhou, China).

Six to eight-week-old male C57BL/6 mice were used to construct bile duct ligation (BDL) and bile drainage (BD) models. The detailed procedures for BDL and BD were performed as described in our previous study [44]. BDL brief procedure: ligation of proximal and distal common bile duct was performed, and then the common bile duct was cut. BD brief procedure: the common bile duct was opened, sterile drainage tube was inserted and reinforced with sutures, and the end of the drainage tube was extracted from the skin of the back of the mice.

### 4.2. Mice Small Intestinal Crypt Isolation

The mice’s small intestines were cut from the terminal ileum up to approximately 15–20 cm, flushed with cold PBS, and opened, and villi with mucus were scraped by using a glass slide. The intestine was cut into small fragments approximately 3–5 mm in length and incubated in 3 mM EDTA solution on ice for 30 min. After removing EDTA solution, crypts were blown down with cold PBS and filtered through 70 μm cell mesh. Crypts were collected by centrifugation at 200× *g* for 5 min.

### 4.3. Organoid Culture

Crypts were resuspended in IntestiCult^TM^ Organoid Growth Medium (STEMCELL Technologies, Vancouver, Canada), plated (approximately 50–100 crypts per 40 μL drop of 50% Matrigel), and overlaid with IntestiCult^TM^ Organoid Growth Medium. Then, the organoids were maintained in a cell incubator at 37 °C containing 5% CO_2_.

### 4.4. Cell line and Cell Culture

The human colon adenocarcinoma cell line Caco-2 was used in this study and obtained from the China Center for Type Culture Collection (CCTCC). Caco-2 cells were cultured in DMEM (Biological Industries, Beit-Haemek, Israel) supplemented with 10% fetal bovine serum (Gibco, Vacaville, CA, USA) and 1% MEM nonessential amino acids (Gibco, Vacaville, CA, USA) and incubated in a cell incubator at 37 °C containing 5% CO_2_.

### 4.5. Chemicals

Chenodeoxycholic acid (CDCA, HY-76847), obeticholic acid (OCA, HY-12222), and corticosterone (CORT, HY-B1618) were obtained from MedChemExpress (Shanghai, China). Dimethyl sulfoxide (DMSO) was obtained from Sangon Biotech (Shanghai, China). Mice were gavaged daily with 10 mg kg^−1^ OCA, 2 mg kg^−1^ CORT, or stroke-physiological saline solution as a control after BDL. Intestinal organoids or Caco-2 cells were treated with 50 μM CDCA, 100 nM OCA, 50 nM CORT, or DMSO as a control.

### 4.6. Serum LPS Detection

A Chromogenic LAL Endotoxin Assay Kit (GenScript, Nanjing, China) was used to detect serum LPS. The protocol can be viewed on the website (https://www.genscript.com.cn/, accessed on 23 Augest 2023).

### 4.7. Statistical Analysis

The data are shown as the means and standard errors and were analyzed via GraphPad Prism 6.0 (GraphPad Software, San Diego, CA, USA). Statistical analysis was expressed with Student’s *t* test, one-way ANOVA, or two-way ANOVA, and differences were considered statistically significant at a level of *p* < 0.05.

Other materials and methods are provided in the Appendix A.

## Figures and Tables

**Figure 1 ijms-24-13494-f001:**
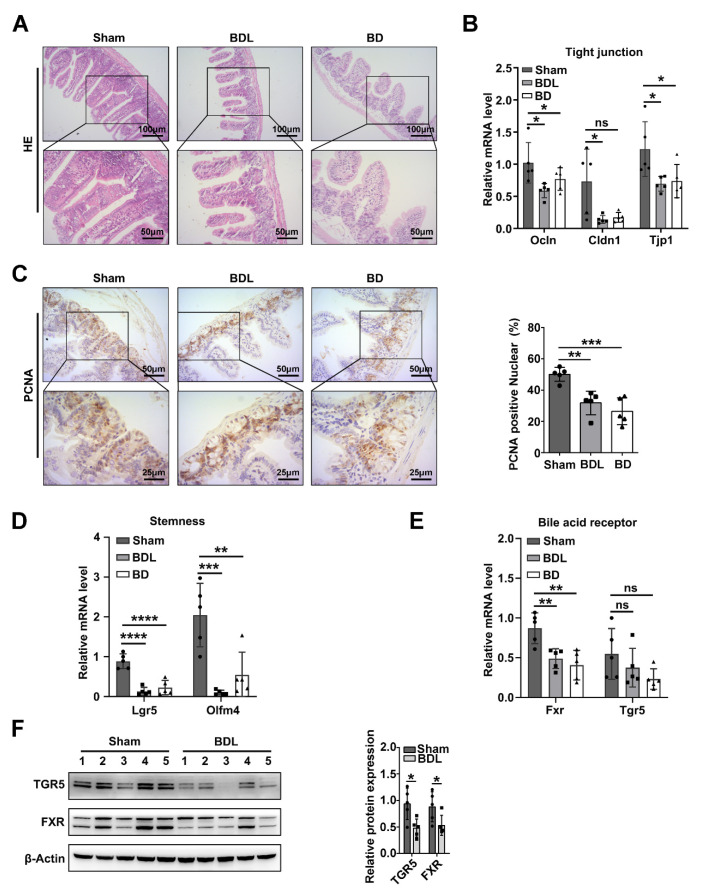
Intestinal bile acid deficiency induced intestinal barrier and stemness and reduced bile acid receptor expression. (**A**) Intestine HE staining of sham, BDL, and BD mice; upper photos with 100× magnification and lower photos with 200× magnification. (**B**) Relative mRNA levels of the tight junction genes *Ocln*, *Cldn1*, and *Tjp1* in different mice groups, *Actb* was used as a reference gene, *n* = 5 per group. Data are shown as the mean ± SD per group, and statistical analysis was performed via one-way ANOVA: * *p* < 0.05. ns: no significant. (**C**) intestine PCNA IHC staining of sham, BDL, and BD mice; upper photos with a 200× magnification and lower photos with a 400× magnification. The quantification of PCNA positive nuclear staining is followed, and statistical analysis was performed via one-way ANOVA: ** *p* < 0.01 and *** *p* < 0.005. (**D**) Relative mRNA levels of the stemness genes Olfm4 and Lgr5 in different mouse groups. Actb was used as a reference gene, *n* = 5 per group. Data are shown as the mean ± SD per group, and statistical analysis was performed via one-way ANOVA: ** *p* < 0.01, *** *p* < 0.005 and **** *p* < 0.001. (**E**) mRNA levels of the bile acid receptors Fxr and Tgr5 in different mice groups, Actb was used as a reference gene, *n* = 5 per group. Data are shown as the mean ± SD per group, and statistical analysis was performed via one-way ANOVA: ** *p* < 0.01. ns: no significant. (**F**) Western blot analysis of FXR and TGR5 in sham and BDL mice intestines. The quantification of TGR5 and FXR protein expression is followed. Data are shown as the mean ± SD per group, and statistical analysis was performed via Student’s *t* test: * *p* < 0.05.

**Figure 2 ijms-24-13494-f002:**
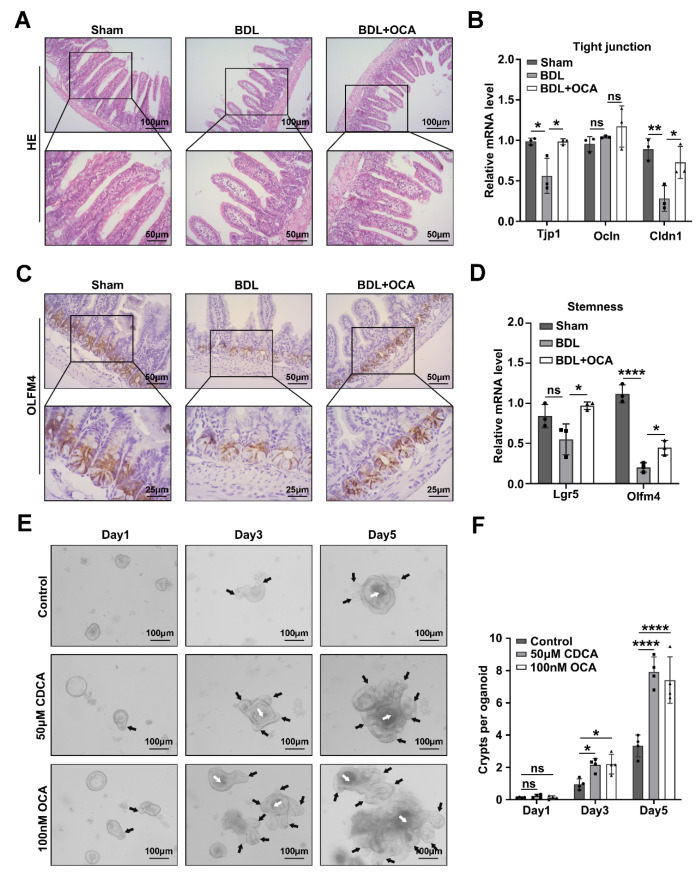
The function of Fxr in maintaining intestinal tight junctions and stemness. (**A**) Intestinal HE staining of sham, BDL, and BDL+OCA mice; upper photos are shown at 100× magnification, and lower photos are shown at 200× magnification. (**B**) Relative mRNA levels of the tight junction genes *Ocln*, *Cldn1*, and *Tjp1* in different mice groups, *Actb* was used as a reference gene, *n* = 3 per group. Data are shown as the mean ± SD per group, and statistical analysis was performed via one-way ANOVA: * *p* < 0.05 and ** *p* < 0.01. ns: no significant. (**C**) Intestine OLFM4 IHC staining of Sham, BDL, and BDL+OCA mice; upper photos with a 200× magnification and lower photos with a 400× magnification. (**D**) Relative mRNA levels of the stemness genes *Olfm4* and *Lgr5* in different mice groups. *Actb* was used as a reference gene; *n* = 3 per group. Data are shown as the mean ± SD per group, and statistical analysis was performed via one-way ANOVA: * *p* < 0.05 and **** *p* < 0.001. ns: no significant. (**E**) The intestine organoids were cultured with 50 μM CDCA, 100 nM OCA, or DMSO as a control, and the development of organoids on day 1, day 3, and day 5 was recorded at 100× magnification. The white arrow indicates the lumen of the organoid, and the black arrow indicates the newborn crypts. (**F**) Quantification of the number of crypts per organoid in these three groups on days 1, 3, and 5. Data are shown as the mean ± SD per group, and statistical analysis was performed via two-way ANOVA: * *p* < 0.05 and **** *p* < 0.001. ns: no significant.

**Figure 3 ijms-24-13494-f003:**
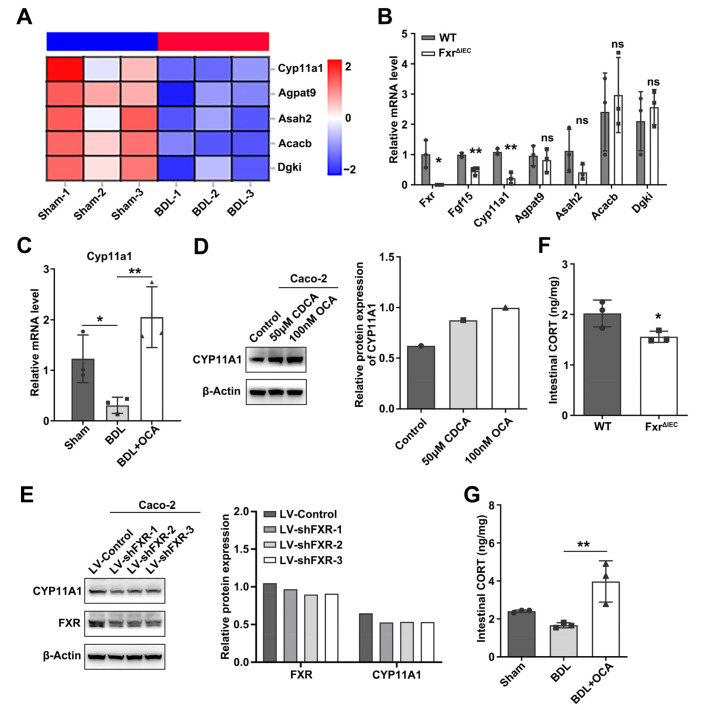
FXR regulates CYP11A1 expression, promoting CORT synthesis. (**A**) Heatmap of the top 5 downregulated lipid metabolism-related genes in BDL mouse intestinal crypts. (**B**) Relative mRNA levels of the top 5 downregulated lipid metabolism-related genes and the Fxr-regulated gene *Fgf15* in *Fxr*^ΔIEC^ and wild-type mice intestine crypts. *Actb* was used as a reference gene, *n* = 3 per group. Data represent the mean ± SD, and statistical analysis was performed via Student’s *t* test: * *p* < 0.05 and ** *p* < 0.01. ns: no significant. (**C**) Relative mRNA levels of the tight junction gene *Cyp11a1* in the sham, BDL, and BDL+OCA mice groups, *Actb* was used as a reference gene, *n* = 3 per group. Data represent the mean ± SD, and statistical analysis was performed via one-way ANOVA: * *p* < 0.05, ** *p* < 0.01. (**D**) The protein expression of CYP11A1 in Caco-2 cells after treatment with 50 μM CDCA or 100 nM OCA or DMSO as a control. (**E**) CYP11A1 protein expression in Caco-2 cells after FXR knockdown. (**F**) The concentration of intestinal CORT in wild-type and *Fxr*^ΔIEC^ mice by ELISA, *n* = 3 per group. Data represent the mean ± SD, and statistical analysis was performed via Student’s *t* test: * *p* < 0.05. (**G**) The concentration of intestinal CORT in sham, BDL, and BDL+OCA mice by ELISA, *n* = 3 per group. Data represent the mean ± SD, and statistical analysis was performed via one-way ANOVA: ** *p* < 0.01.

**Figure 4 ijms-24-13494-f004:**
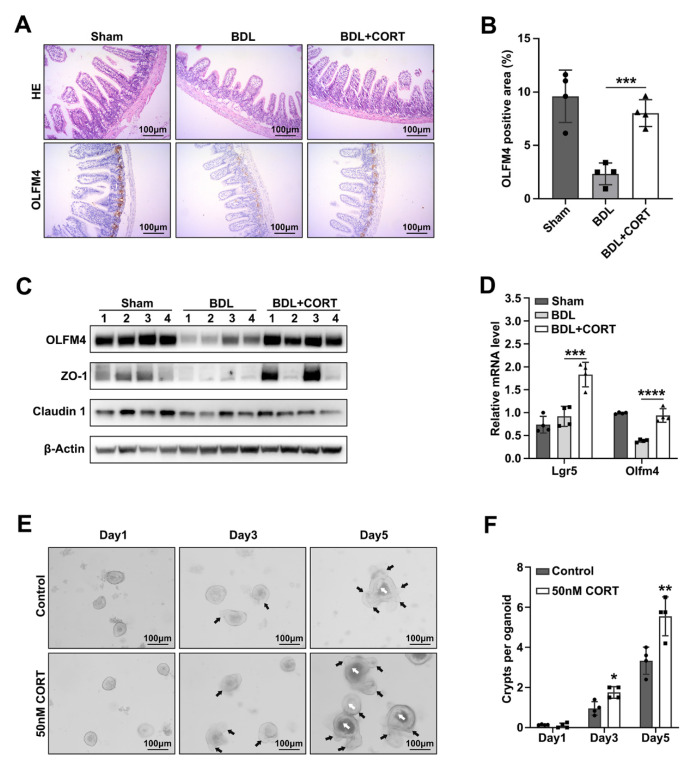
CORT protects the intestinal barrier and stemness after BDL. (**A**) HE staining and OLFM4 IHC staining of intestine in Sham, BDL, and BDL+CORT mice group, with a 100× magnification. (**B**) Quantification of the OLFM4-positive area, *n* = 4 per group. Data are shown as the mean ± SD per group, and statistical analysis was performed via one-way ANOVA: *** *p* < 0.005. (**C**) The protein expression of OLFM4, ZO-1, and Claudin 1 in sham, BDL, and BDL+CORT mice intestines. (**D**) Relative mRNA levels of *Lgr5* and *Olfm4* in sham, BDL, and BDL+CORT mice intestines. Actb was used as a reference gene, *n* = 4 per group. Data are shown as the mean ± SD per group, and statistical analysis was performed via one-way ANOVA: *** *p* < 0.005 and **** *p* < 0.001. (**E**) The intestine organoids were cultured with 50 nM CORT or DMSO as a control, and the development of organoids on day 1, day 3, and day 5 was recorded at 100× magnification. The white arrow indicates the lumen of the organoid, and the black arrow indicates the newborn crypts. (**F**) Quantification of the number of crypts per organoid in these three groups on days 1, 3, and 5. Data are shown as the mean ± SD per group, and statistical analysis was performed via two-way ANOVA: * *p* < 0.05 and ** *p* < 0.01.

**Figure 5 ijms-24-13494-f005:**
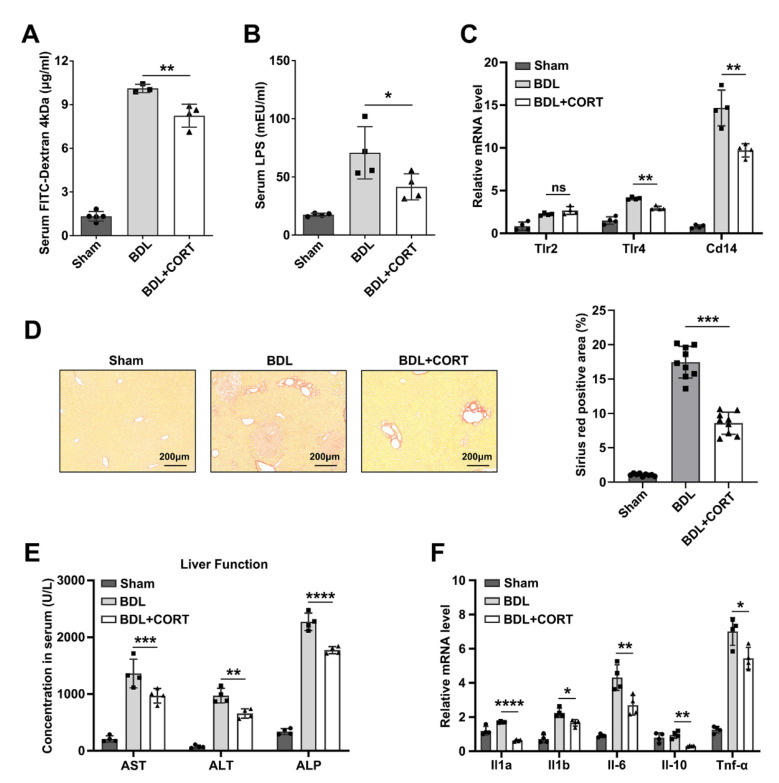
CORT alleviates liver injury by decreasing intestinal permeability. (**A**) Concentration of serum FITC-dextran 4 kDa to assess the change in intestinal permeability, *n* = 4 per group. Data are shown as the mean ± SD from each group. Statistical analysis was performed via one-way ANOVA: ** *p* < 0.01. (**B**) Concentration of serum LPS in the sham, BDL, and BDL+CORT mice groups, *n* = 4 per group. Data are shown as the mean ± SD from each group. Statistical analysis was performed via one-way ANOVA: * *p* < 0.05. (**C**) Relative mRNA levels of LPS-related genes in sham, BDL, and BDL+CORT mice livers. *Actb* was used as a reference gene, *n* = 4 per group. Data are shown as the mean ± SD from each group. Statistical analysis was performed via one-way ANOVA: ** *p* < 0.01. ns: no significant. (**D**) Sirius Red staining of sham, BDL, and BDL+CORT mouse livers and quantification of the Sirius Red-positive area, *n* = 4 per group. Data are shown as the mean ± SD from each group. Statistical analysis was performed via one-way ANOVA: *** *p* < 0.005. (**E**) Serum concentrations of ALT, AST, and ALP in sham, BDL, and BDL+CORT mice, *n* = 4 per group. Data are shown as the mean ± SD from each group. Statistical analysis was performed via one-way ANOVA: ** *p* < 0.01, *** *p* < 0.005 and **** *p* < 0.001. (**F**) Relative mRNA levels of inflammatory cytokines in sham, BDL, and BDL+CORT mice livers. *Actb* was used as a reference gene, *n* = 4 per group. Data are shown as the mean ± SD from each group. Statistical analysis was performed via one-way ANOVA: * *p* < 0.05, ** *p* < 0.01, and **** *p* < 0.001. ns: no significant.

## Data Availability

Data will be made available upon request.

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
