# Peer review of "FXR Maintains the Intestinal Barrier and Stemness by Regulating CYP11A1-Mediated Corticosterone Synthesis in Biliary Obstruction Diseases"

_ijms, 2023, doi:10.3390/ijms241713494_

Round 1

Reviewer 1 Report

This manuscript investigates the roles of the farnesoid X receptor (FRX) and bile acids in regulation of the intestinal epithelial barrier and mechanisms underlying gut barrier disruption during bile duct obstruction. The manuscript targets important scientific topic and presents some interesting data. Yet the present paper contains a number of problems that need to be addressed.  

Comments:

1. Introduction is poorly written and contains rather primitive and incomplete disruption of epithelial apical junctions and their roles in regulation of the intestinal epithelial barrier. 

2. Materials and Methods section is unacceptable. Description of several important techniques is missing. Examples include lack of description of immunoblotting, immunohistochemistry, RT-PCR and shRNA-mediated gene knockdown. All important reagents and their sources should be also included. Finally, more information should be provided for Nrlh4 conditional knockout mice: their background, housing conditions (co-housing of control and knockout mice) and references.

3. The authors should include a brief description of the BDL and BD models.

4. Lines 174-175 have an erroneous statement that the roles of bile in gut barrier disruption during biliary obstruction is unknown. Please modify this statement.

5. Figure 1C. PCNA staining in animal intestinal sections should be quantified.

6. Figure 1F. Effects of BDL on expression of TGR5 and FXR are variable, therefore the blots should be subjected densitometric quantification.

7. Figure 2E,F. It is unclear if the authors used two different concentrations of CDCA or CDCA and OCA in the described experiments. They need to explain why OCA was used in animal studies whereas CDCA was used to treat ex vivo mouse organoids.

8. Figure 3F is confusing. It is unclear if the authors achieved significant knoc-down of FXR in Caco-2 cells using their shRNAs and whether such knockdown had any functional effect, such as decreasing CYP11A1 expression.

Reviewer 2 Report

The authors try to explain the way to prevent intestinal complications and alleviation of liver injury after biliary obstruction by focusing on the intestinal barrier functions. For this purpose, to investigate the mechanism by which intestinal bile acid deficiency mediates intestinal barrier dysfunction after biliary obstruction, they developed a  bile duct ligation (BDL) mice model and studied the role of the bile acid receptor FXR in maintaining intestinal barrier function. By advanced techniques, such as RNA-seq analysis of BDL and sham mice crypts and qRT‒PCR, they found that FXR might maintain intestinal stemness by regulating CYP11A1 expression responsible for glucocorticoid production. In this vein, they found that FXR activation could promote intestinal corticosterone synthesis.

Their study shows the relation between intestinal barrier function and bile obstruction, a statement that opens new therapeutical approaches.

It is a well-written scientific manuscript that gives new information on the therapeutical approach. The paper should be of high interest to readers and pharmaceutical companies. Their study is well designed based on an in-depth molecular and laboratory analysis for all different sides.

Results are consistent followed by an in-depth discussion based on bibliographical data.

The paper merits publication in its present form.

Author Response

Thanks for your recognition of our study. And we will also conduct further research on this study.

Round 2

Reviewer 1 Report

The authors did a good job in addressing my comments and questions. I have nothing to add.